# Hop Latent Viroid: A Hidden Threat to the Cannabis Industry

**DOI:** 10.3390/v15030681

**Published:** 2023-03-04

**Authors:** Charith Raj Adkar-Purushothama, Teruo Sano, Jean-Pierre Perreault

**Affiliations:** 1RNA Group, Department of Biochemistry and Functional Genomics, Université de Sherbrooke, Sherbrooke, PQ J1E 4K8, Canada; 2Faculty of Agriculture and Life Science, Hirosaki University, Hirosaki 036-8561, Japan

**Keywords:** hop latent viroid, cannabis disease, HLVd, viroid, duds, dudding, cannabis pathogen, hop viroid, viroid disease

## Abstract

Hop latent viroid (HLVd) is the biggest concern for cannabis and hop growers worldwide. Although most HLVd-infected plants remain asymptomatic, research on hops has demonstrated a decrease in both the α-bitter acid and terpene content of hop cones, which affects their economic value. The HLVd-associated “dudding” or “duds” disease of cannabis was first reported in 2019 in California. Since then, the disease has become widespread in cannabis-growing facilities across North America. Although severe yield loss associated with duds disease has been recorded, little scientific information is available to growers in order to contain HLVd. Consequently, this review aims to summarise all of the scientific information available on HLVd so as to be able to understand the effect of HLVd on yield loss, cannabinoid content, terpene profile, disease management and inform crop protection strategies.

## 1. Introduction

Since its initial detection in 2019 in California, the “duds” disease of cannabis has become the most devastating cannabis disease (syn. Hemp) (*Cannabis sativa*, *Cannabis indica*, *Cannabis ruderalis*) in cannabis-growing areas. The causative agent of this disease was found to be hop latent viroid (HLVd). A survey conducted in 2021 by the Dark Heart Nursery Research that involved 200,000 tissue tests concluded that 90% of cannabis--growing facilities in California were contaminated with HLVd. Furthermore, the researchers noted that the HLVd infection could significantly affect the plants’ vigor and yield. It has been estimated that it could cause losses of up to USD 4 billion annually for the cannabis industry (https://www.cannabisbusinesstimes.com/article/cannabis-hop-latent-viroid-infections-dark-heart-nursery-crop-loss/ [accessed on 13 February 2023]). HLVd has been detected throughout the cannabis-growing facilities of North America (https://nova-analyticlabs.com/nova-labs-tests-hlvd-and-two-other-diseases/ [accessed on 13 February 2023] https://hempindustrydaily.com/this-elusive-pathogen-is-damaging-hemp-nationwide-heres-how-to-fight-it/?cn-reloaded=1 [accessed on 13 February 2023]). Another report indicated that infected crops could suffer anywhere from a 50%–70% loss in tetrahydrocannabinol (THC) content, thus considerably lowering their commercial value (https://stratcann.com/2021/04/01/the-hop-latent-viroids-warning-shot-to-the-canadian-cannabis-industry/ [accessed on 13 February 2023]).

## 2. Viroids

To date, viroids are the smallest known infectious agents that are predominantly known to cause disease in plants. The term “viroid” was proposed by Theodor Diener in 1971 to describe a protein-free, small (50–80 times smaller than the smallest viral genomes), low molecular weight, plant pathogenic RNA molecule [1]. The potato spindle tuber viroid (PSTVd) was the first viroid species to be identified and characterized [2]. Specifically, PSTVd was isolated from Irish potatoes showing degenerative disease in North America. The existence of “viroids” was further supported by the subsequent discovery of citrus exocortis viroid (CEVd) from citrus plants exhibiting citrus exocortis disease [3]. Since then, more than 50 viroid/viroid-like RNAs have been isolated from different host plants, including both mono- and dicotyledons. Advancements in plant pathogen diagnosis technology, new viroids and their distribution in new geographic areas are constantly being reported [4]. Viroid disease symptoms in the host plant greatly depend on both the viroid variant and the host plant cultivar. For instance, the PSTVd variant intermediate (PSTVd-I GenBank accession No. AY937179) induces severe disease symptoms in tomato cultivar Rutgers, but it is asymptomatic in tomato cultivar Moneymaker, even though all plants were cultivated under identical environmental conditions. On the other hand, two viroid variants of the same viroid species can induce different disease symptoms on the same plant cultivar. For example, PSTVd-I and PSTVd-RG1 (GenBank accession No. U23058), which differ by only three nucleotides, induce intermediate and severe disease symptoms on tomato cultivar Rutgers, respectively.

To date, 44 species of viroids have been characterized and classified into either the *Avsunviroidae* or the *Pospiviroidae* families based on the structure of the mature viroid RNA, the central conserved region (CCR), any self-cleaving activity (i.e., ribozyme), the subcellular site of replication in the host plant, the mode of replication and the enzymes involved in that replication [5]. There are five members in the family *Avsunviroidae*. All of these viroid species replicate in chloroplasts through a symmetric rolling-circle mechanism and exhibit self-cleavage activity. All of the remaining 39 viroid species are grouped under the family *Pospiviroidae*. Members of the *Pospiviroidae* family replicate in the host’s nucleus through an asymmetric rolling-circle mechanism and are characterized by the presence of five structural/functional domains (the terminal left (TL), pathogenicity (P), central (C), variable (V) and terminal right (TR) domains).

## 3. Hop Latent Viroid

In 1987, Pallas et al. reported the presence of viroid-like RNA in nucleic acid preparations from two of the three commercial hop varieties (*Humulus lupulus* L., family, *Cannabaceae*) grown in the León region of Spain. Although this RNA molecule had the same size as the avocado sunblotch viroid (ASBVd), its physical and biological properties differed from those of all previously characterized viroids [6]. Since this viroid-like RNA showed a faster electrophoretic mobility than the hop stunt viroid (HSVd; 297-nt in size), it was tentatively identified as “hop viroid-like RNA fast” (HV-f). In order to evaluate the possible infection of hops by HV-f, a survey was conducted in 17,000 hectares of hop gardens in Hallertau, Germany, the main German hop-growing region. This survey revealed the presence of HV-f in all of the hop cultivars examined [7]. Since HV-f did not induce any visible disease symptoms in hops, it was tentatively named “hop latent viroid” (HLVd). Comprehensive worldwide surveys revealed the presence of HLVd in most of the hop cultivars tested [7]. Although HLVd-infected hop plants are symptomless, infection significantly reduces both yield and either the α-bitter acid or the essential oil content in the hop cone, thus reducing its market value [8]. Subsequently, HLVd has been detected in 90% to 100% of the tested hop germplasms in European countries [9].

Both biochemical and molecular biology analyses revealed that HLVd is a covalently closed, circular RNA of 256-nt. Thermodynamically, HLVd folds into a rod-like secondary structure, with 65.5% of the residues being internally base-paired (see below). Although HLVd had a CCR, it lacked an oligo (A) stretch in the upper part of the left domain in its secondary structure. HSVd, which is known to cause stunting disease in hops, exhibited 45% sequence similarity with HLVd, while PSTVd was found to be 51% identical with it. The closest sequence identity found with HLVd was coconut cadang-cadang viroid (CCCVd; 54%), while ASBVd (36%) was least similar [7]. Based on structural features such as the rod-like conformation, the presence of the five functional domains, the absence of any self-cleavage activity, the presence of both a terminal conserved hairpin (TCH) and a CCR identical to that of members of the other species of the genus, HLVd was placed in the genus *Cocadviroid* of the family *Pospiviroidae* (Table 1).

## 4. Host Range and Transmission of HLVd

Until mid-2019, HLVd was known to infect only three natural hosts, specifically the commercial hop (*Humulus lupulus*), the Japanese hop (*Humulus japonicus* Sieb. and Zucc.) (Figure 1) and the stinging nettle (*Urtica dioica* L.) [10]. Heat-generated HLVd variants (i.e., thermomutants) can infect what otherwise are considered as non-susceptible species such as tomato (*Solanum lycopersicum*) and *Nicotiana benthamiana* [11]. In 2019, two independent groups reported the detection of HLVd in stunted cannabis plants in the USA [12,13].

With the globalization of agriculture, viroids have become widely distributed in both new environments and new geographical areas. For example, HLVd was introduced to Brazil from the USA through hop germplasm [14]. HLVd can be transmitted over long distances and can be introduced into hop gardens by infected propagative materials. HLVd transmits within the hop yards mechanically, by grafting and vegetative propagation, and by contaminated tools or machinery [15]. HLVd transmission by either pollen or seed has been reported as being either low or non-existing, respectively [10,16]. Despite the new infections away from the infected hop yards are common, to date insect transmission of HLVd is not known [17].

## 5. HLVd-Associated Symptoms in Hops

Although HLVd-infected hop plants are symptomless, infection significantly reduces both cone yield and either the bitter acid or the essential oil content in the hop cones of sensitive cultivars [8]. Currently, HLVd has been reported in commercial hop yards in Australia, New Zealand, China, Japan, Republic of Korea, Russia, Slovenia, Poland, Germany, the Czech Republic, Hungary, France, Spain, Portugal, the United Kingdom, South Africa, the USA and Brazil [15]. However, due to the asymptomatic nature of the HLVd infection in most of these hop cultivars, these HLVd infections have long gone unnoticed. HLVd symptoms are evident only by comparison of infected susceptible and healthy hop cultivars. In susceptible cultivars such as ‘Omega’, HLVd-associated symptoms include chlorosis, slow growth and the presence of fewer and smaller cones [8]. HLVd also affects both nursery production and breeding programs since it significantly reduces both the rooting and the establishment of softwood cuttings [10].

## 6. Effect of HLVd on the Terpene and Essential Oil Contents

Despite the absence of any characteristic symptoms in somatic tissues, HLVd induces physiological changes that affect both qualitatively and quantitatively the metabolites of the lupulin secretory glands and of the essential oils, thus indicating that the HLVd infection is not a truly latent one [18]. In HLVd-infected plants, the yield loss can be as low as 8% for the Wye Challenger cultivar, and as high as 37.5% for the Slovenian hop cultivar [8,19]. The reductions in α-bitter acid were 15 and 30% for the cultivars Wye Challenger and Omega, respectively. Both cultivars showed increased β-bitter acid and oil contents [18]. A higher amount of β-bitter acid and oil content leads to the early maturation of the hop cones [8]. The effect of HLVd on the α-bitter acid content was genotype-dependent. This reduction in α-bitter acids ranged from 20% to 50% within English hop cultivars [17]. Other hop cultivars in the Czech Republic also showed signification reductions in the α-bitter acid content in HLVd-infected plants (ex. Saaz, 40% reduction; Premiant, 40% reduction; Aurora, 18% reduction; and, Sybilla, Marynka, Pulawski and Magnat, from 11% to 23% reduction) [19].

HLVd is also known to affect both the oil and the terpene profiles of hops. For instance, HLVd infection increased the levels of monoterpenes such as myrcene and both α- and β-pinene by 29% to 41.6% as compared to healthy plants [20]. On the contrary, HLVd infection decreased the levels of sesquiterpenes such as β-caryophyllene, α-humulene, α-copaene, γ-muurolene, β-bisabolene, γ-cadinene and δ-cadinene by 13 to 29%. The possible influence of some oxidative-reduction processes that are activated by the viroid-caused pathogenesis was assumed to be involved in the accumulation of terpenes alcohols such as geraniol and methylgeranate, as well as in the reduction in the levels of the majority of the ketones detected in the spectra of the essential oils [20]. However, these changes in the composition of the essential oils present in the hop cones are both genotype-dependent and cultivar-specific [21]. For instance, the linalool content in the cones of infected plants was found to be higher for cultivars Sybillla, Lubelski and Pulawski, but lower for cultivars Marynka and Magnat. The methylgeranate content was found to be lower in the infected plants from all cultivars [20]. In terms of the importance of the HLVd infection of hops with economic importance, even the slightest differences in either the content or composition in α-bitter acid, β-bitter acid, oil content and in the terpene profile can change the resulting beer’s aroma [10]

## 7. HLVd Disease in Cannabis

HLVd disease in Cannabis plants is loosely described as “duds” or “dudding disease” (https://stratcann.com/insight/the-hop-latent-viroids-warning-shot-to-the-canadian-cannabis-industry/ [accessed on 13 February 2023]). Like in hops, only a few cultivars of cannabis show HLVd-associated symptoms, implying that both symptom expression and disease severity are cultivar genotype dependent. In 2014, cannabis growers started an on-line thread discussing the “dudding disease” symptoms in cannabis plants (https://www.thcfarmer.com/threads/what-to-do-with-duds.64342/ [accessed on 13 February 2023]). The association of HLVd infection with symptomatic plants was confirmed in 2019 by two independent teams using high-throughput sequencing technology and subsequent bioassays [12,13]. In susceptible cultivars HLVd induces symptoms (Figure 2) such as shorter internodal spacing, smaller leaves, stunting, malformation (outwardly horizontal plant structure), chlorosis, brittle stems, reduced vigor, lower water intake, reduced flower mass and trichomes [12,13]. At the flowering stage, susceptible plants typically show smaller and looser buds, weaker flower smell and less trichome production. This effect is reflected in both the yield and loss of quality that includes up to a 50% reduction in both cannabinoid and terpene production (https://www.plantcelltechnology.com/blog/everything-you-should-know-about-hop-latent-viroid-hplvd/ [accessed on 13 February 2023]). A survey conducted in 2021 revealed that approximately 90% of all cannabis-growing facilities in California tested positive for HLVd, and 30% of the plants in each facility showed symptoms of the viroid’s infection (https://docs.google.com/document/d/1G5Lwz14F5-baVGveMe_0yGnam7z1E0qQZJ8YAcardFw/edit# [accessed on 13 February 2023]). Since its first detection in California, HLVd did not take much time to find its way to British Columbia, Canada. Now, HLVd is prevalent in Canadian cannabis-growing facilities. This clearly illustrates the severity of HLVd disease and its threat to the cannabis industry in North America.

## 8. Comparison between HLVd Isolated from Hops and Cannabis

Due to the absence of a protein coat around the viroid genome, its secondary structure plays a significant role in determining its ability to invade the host plant, survival and pathogenesis. Therefore, understanding the secondary structure of HLVd is paramount to understanding the host-viroid relationship. Due to the non-coding nature of viroids, they recruit a host DNA-dependent RNA polymerase during replication. Specifically, nucleus-replicating viroids such as HLVd use DNA-dependent RNA polymerase II [22]. Since it is an abnormal condition for DNA-dependent RNA polymerase II to use RNA as a template, the resulting replication is error-prone [23]. The sequence variants created by the replication of a master sequence during this process are called “quasi-species” [24]. Analysis of high-throughput sequencing data obtained from PSTVd-infected plants revealed the presence of “quasi-species” of the members of the family *Pospiviroidae* [25,26]. However, the sequence analysis of HLVd-infected hop plants revealed only a small number of sequence variants as compared to what has been seen with other viroid species [7,27].

In order to understand both the sequence variation and adaptation of HLVd to cannabis plants, the HLVd-type species isolated from hops (GenBank Acc. No.: NC_003611) and all of the HLVd sequences isolated from cannabis plants were compared. In order to achieve this goal, all of the HLVd sequences isolated from cannabis plants that are available in NCBI were retrieved, and only the full-length sequence (256-nt) was considered for the analysis. Warren et al. [13] reported two distinct HLVd isolates, specifically isolates Can1 (GenBank Acc. No.: MK876285) and Can2 (GenBank Acc. No.: MK876286). The Can1 isolate showed a 100% sequence similarity with the HLVd-type species, while the Can2 isolate had one mismatch. Specifically, at nucleotoide 225, the uracil (U) was mutated to adenine (A) in isolate Can2 as compared to the HLVd-type species. Hereafter, this point mutation will be referred to as U225A. However, the sequence of the Can2 isolate was 100% identical to an HLVd isolate retrieved from a commercial hop garden in China (GenBank Acc. No.: EF613183). Interestingly, both isolates have also been reported from HLVd-infected cannabis plants located elsewhere in the USA. More specifically, HLVd reported from Delta county (CO, USA) was found to be 100% similar to the Can1 isolate, and the Can2 isolate matched with an HLVd isolated from both Santa Barbara (CA, USA) and Boulder counties (CO, USA) [12,28]. This indicates the presence of at least two HLVd sequence variants infecting cannabis.

In order to understand the effect of U225A on HLVd structure, the secondary structure of the HLVd Can2 isolate was predicted using the RNAfold WebServer (http://rna.tbi.univie.ac.at/cgi-bin/RNAWebSuite/RNAfold.cgi [accessed on 30 November 2022]) and was compared with the secondary structure of the HLVd-type species. The ΔG values for both the Can2 isolate and the HLVd-type species were found to be −95.30. The single nucleotide change (U225A) did not affect the structure of the Can2 isolate as compared to that of the type species. Although a single point mutation at position 225 did not alter the secondary structure, it is interesting to note that this change is located within the lower pathogenicity domain (Figure 3). Previously, we have isolated a variant of PSTVd from Dahlia (PSTVd-D; GenBank Acc. No.: AB623143) that induces mild symptoms on tomato cultivar Rutgers as compared to what has been seen with PSTVd-I. At the genomic level, these two differed by nine nucleotides [29]. Through mutagenic studies, it was demonstrated that a change in one nucleotide in the pathogenicity domain of PSTVd-D is crucial in both disease symptoms attenuation and in the escape from the host’s defense mechanism in the tomato cultivar Rutgers [30]. Similarly, the HSVd isolated from hops (HSVd-hop type) adapts to grapevines by the presence of five point mutations [31]. Hence, it would be interesting to study the effect of the single nucleotide change observed in the pathogenicity domain of the HLVd isolated from cannabis on the host’s transcriptome, on the disease’s severity and on its adaptability to the host plant.

## 9. Control of HLVd Disease

The basis of viroid control is the production of viroid-free propagative materials. There are several layers of control measures that are involved. The first and foremost is the prevention provided by testing all of the incoming cannabis plants and cannabis products that could potentially act as a carrier for HLVd. Hence, it is good practice to test the plants and products for potential HLVd contaminants before bringing them into the facility. Once the plants are received, a 30-day quarantine is critical in order to properly evaluate for the presence of plant pathogens in incoming plant material. Given that HLVd infection is asymptomatic in many cannabis cultivars, it is recommended to test any new plant during the third week of quarantine, and certainly before sending the newly received plants into production. Although there is no chemical treatment available for controlling HLVd disease, methods such as meristem tip culture [32], thermotherapy [33] and cold treatment [34] have been found to reduce viroid titer significantly. Meristem culturing of shoot tip successfully eliminated HLVd from two infected hop cultivars [35]. However, combining cold treatment (incubating plant materials at 2 to 4 °C in the dark for periods ranging from 8 to 21 months) and the meristem culture of tips of less than 0.5 mm in size was found to be more effective in eliminating HLVd [34]. Thermotherapy involving incubating plants at 36 °C for 14 days rapidly decreased viroid titer, while the original titer was restored after six months under field conditions. The decrease in HLVd titer was correlated with the induction of an RNA silencing mechanism that resulted in the cleaving of the viroid RNA [33]. Unlike other viroid species, HLVd is known to produce negligible levels of mutated variants under standard cultivation conditions. However, thermotherapy resulted in a significant increase in the number of sequence variants detected in the HLVd population. All mutated cDNAs were infectious and evolved into complex progeny populations that contained low levels of the molecular variants. These thermomutants were found to be infectious to both tomato and *Nicotiana benthamiana* plants, both of which are a non-host for the HLVd-type species [18]. Although both cold and heat treatments could potentially reduce HLVd titer, subsequent meristem tissues are required to completely eliminate all of the viroid [10].

## 10. Detection and Management of HLVd Infection

As discussed elsewhere, HLVd primarily spreads via vegetative cuttings, contaminated tools and by other mechanical means within either the hop yards or the indoor cannabis-growing facility. Since there are no control measures available for viroid-associated diseases, HLVd spread can only be prevented by employing a multilayer management system that includes, but is not limited to, the timely identification and removal of infected plants. The absence of visible symptoms associated with HLVd challenges the ability of growers to contain these viroids. Biological indexing is not practical for HLVd due to its limited host range and lack of symptomatology. However, if a susceptible host plant is available, graft inoculation was found to be more effective than rub-inoculation for the infection of hop plants with the hop latent viroid [36]. Due to these difficulties, and to the laborious procedure of biological indexing, molecular detection methods such as dot-blot hybridization, RT-PCR and RT-qPCR can be employed for routine diagnosis [27]. Since viroid distribution is uneven in the host plant, it is important to take multiple leaf samples from the lower to the upper stem, covering both old and new leaves. Although partial replanting following the removal of diseased and adjacent plants can be effective, better control has been achieved by removing all plants in a yard, followed by replanting with certified viroid-free material [10]. Roguing for viroid-infected hops required that all plants, including rhizomes, be destroyed by treating with urea and chloropicrin in the autumn in order to ensure that the disease is not carried over to the succeeding crop [10]. However, in the case of cannabis (which is grown indoors), incineration or treating with bleach could effectively eliminate the liberation of HLVd into the environment. Studies on HSVd demonstrated that it could be found in hop residue; however, leaves and cones were found to lose their infectivity within three months when left to be weather-beaten [37]. That said, HLVd can survive longer in plant material (including the cannabis flower) if it is stored at lower than room temperature. This suggests the chance of HLVd transmission through cannabis products that are either transferred between the producers or are purchased in the market.

The removal of viroid-infected plants, and the sanitation of equipment, tools and of the vicinity of the viroid-infected plants are critical to checking the spread of the viroid in both fields and in indoor growth facilities. Heating blades at 160 °C for 10 min was found to be effective, while doing so at 140 °C was not. These measures have seen the virtual elimination of HSVd from the Kirin Brewery yards in Japan [10]. Among the different heat and chemical treatments tested for tool sanitation against viroids, the most reliable and widely used one is an aqueous solution of at least 5% household bleach (sodium hypochlorite, minimum 1% available chlorine) [15]. However, the treatment of greenhouse tools with 10% regular Clorox bleach (an active ingredient of 5.25% sodium hypochlorite [NaOCl]) for 10 s was found to be effective against the transmission of PSTVd in tomatoes [38]. The strong corrosive effect on both the greenhouse structure and tools, and the potential phytotoxic effect on plants, should be considered before employing bleach as a sanitizer. Another alternative is Virkon S, which has already been proven effective against both human and animal viral pathogens [39,40]. Although it is relatively expensive and corrosive, it is the most promising disinfectant against viroids and viruses infecting greenhouse-grown tomatoes when it is used at a 2% concentration (20 g/L) [38]. In addition to testing and sanitation procedures, it is important to maintain both high standard cultural practices and high personal hygiene practices. This includes, but is not limited to, regular handwashing and the changing of personal protective equipment (PPR) between the hop yards or cultivation rooms.

## 11. Viroid Species Known to Infect Hops

Aside from HLVd, at least three other viroid species are known to infect hops. As hops and cannabis belong to the same family, there is a significant risk that these viroids can also adapt and infect cannabis under ideal environmental conditions. Such a scenario is not uncommon for viroids. For instance, PSTVd is known to cause disease in potato and tomato plants, both of which belong to the family *Solanaceae*. HSVd is one of the very important viroids infecting hops. HSVd was first discovered in Japanese hop fields, with typical symptoms, including stunting, leaf curling, small cone formation and a substantial reduction of alpha-acid content, being observed after 3–7 years of infection, [41]. HSVd has a very broad host range and has been reported to be the causal agent of diseases such as citrus cachexia, cucumber pale fruit, peach and plum dapple fruit and hop stunt [42]. Due to its quasi-species nature, HSVd can quickly adapt to grapevines, which serve as a reservoir for the viroid [43]. HSVd was reported to infect hops and stone fruits in the USA and Canada, respectively [44,45]. The apple fruit crinkle viroid (AFCVd), first identified as the causal agent of apple fruit crinkle disease, induces stunting in hops that resembles the HSVd-associated symptoms observed in Japan [41] and also infects persimmon asymptomatically. Among all hop-infecting viroids, the citrus bark cracking viroid (CBCVd) is the most aggressive, and symptoms appear after one year of infection. In hops, symptoms include leaf down curling, severe bine stunting, a reduction in cone size, dry root rotting after the first dormancy and complete plant dieback in 3–5 years [46]. However, it is interesting to note that the infection of hop by CBCVd alone is not observed in either nature or in hop fields [47]. In other words, CBCVd is found only in hop plants already infected with HLVd. Experimentally, the hop plants co-inoculated with CBCVd and HLVd plants showed more severe disease symptoms than those inoculated with only either CBCVd or HLVd [47]. It should be noted that both HLVd and CBCVd belong to the same viroid genus, namely *Cocadviroid*. Figure 4 illustrates HSVd-, AFCVd- and CBCVd-associated symptoms on hop, respectively.

## 12. Conclusions and Prospective

Although HLVd infection was first reported in hop plants in 1988, both the scientific community and stakeholders prioritized research on HSVd which caused more severe visible disease symptoms in infected hop plants than did HLVd. However, scientific data on the negative effects of HLVd on both the α-bitter acid and terpene contents in both symptomatic and asymptomatic hop collected over the years have attracted the attention of researchers in recent years. The effect of HLVd on asymptomatic hop plants is supported by recent transcriptomic studies conducted on hop plants co-inoculated with HLVd and CBCVd aimed at understanding the increased aggravation of the CBCVd-induced disease symptoms in the presence of HLVd [47]. Although two HLVd sequence variants were detected in cannabis plants, it is not clear whether the identified sequence variants are the result of viroid quasi-species or true dominant HLVd variants (refer to Adkar-Purusothama et al., [26] for details on viroid quasi-species). Hence, further studies are required in order to understand whether or not both of the HLVd sequence variants are able to infect and induce disease symptoms in cannabis plants. If so, what are the HLVd sequence variant-specific disease symptoms on the susceptible cannabis plant and HLVd variant-specific effect on the cannabinoids. These studies will help in understanding the HLVd “master sequence” that cause the initial infection in cannabis plants, and in the study of the HLVd-cannabis host interaction. Additionally, it would be interesting to understand the role of the U225A mutation located in the lower pathogenicity domain in both HLVd’s pathogenicity and its adaptability to cannabis plants. Since HSVd, AFCVd and CBCVd are all infectious to hop plants, it is worth conducting a bioassay on cannabis plants and preparing to contain them all if any infection is found.

To date, HLVd-resistant cannabis cultivars are not known. Meristem tissue culture is the only effective control method via which infected plants can be saved. This process is both laborious and expensive. Although tissue-cultured plantlets are viroid-free, it is important to understand that they are not viroid-resistant. Hence, following preventive measures in order to avoid an insurge of HLVd into a given growth environment is crucial in HLVd-associated disease management. However, for sustainability reasons, it is important to find a practical long-term solution by employing strategies such as the control of HLVd infection by breeding HLVd-resistant plants, by cross-protection and by developing RNA interference-mediated resistance in the plants.

## Figures and Tables

**Figure 1 viruses-15-00681-f001:**
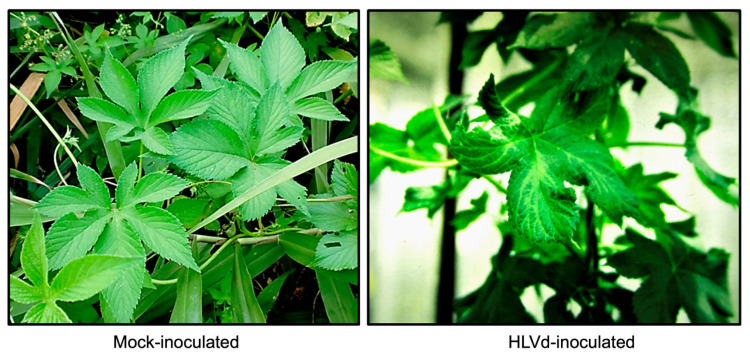
Symptoms (leaf distortion and vein yellowing) on Japanese hop plants infected with HLVd.

**Figure 2 viruses-15-00681-f002:**
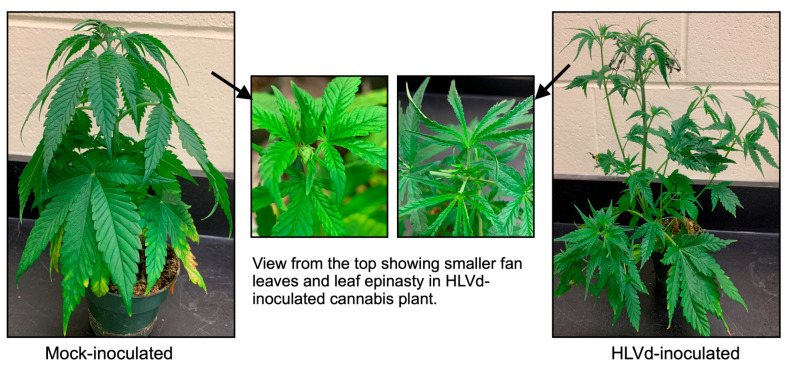
HLVd associated symptoms in susceptible cannabis plants.

**Figure 3 viruses-15-00681-f003:**
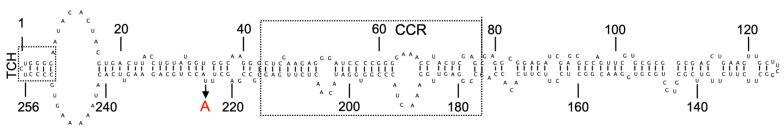
Secondary structure prediction of the HLVd-type species. The boxed regions are the terminal conserved hairpin (TCH) and Central Conserved Regions (CCR). The arrow indicates the U225A point mutation observed in the HLVd isolated from a commercial hop gardens of China.

**Figure 4 viruses-15-00681-f004:**
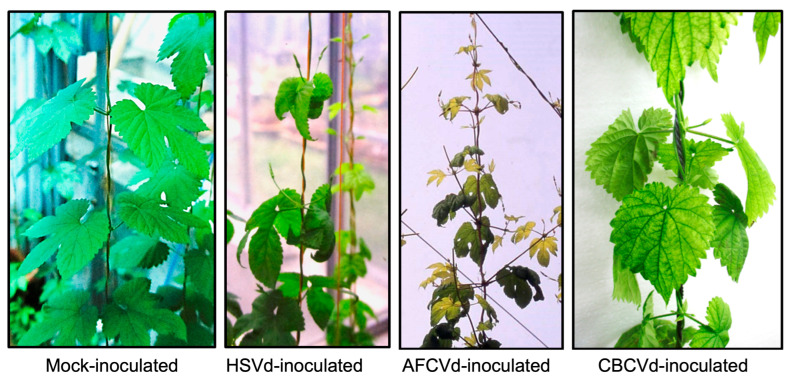
The disease symptoms observed in hop plants infected with HSVd included curled drooping leaf, while those infected with AFCVd showed both shortened internodes and leaf down curling (the yellowing of the leaves is not a symptom, but rather is a cultivar characteristic). Lastly, those infected with CBCVd showed leaf chlorosis.

**Table 1 viruses-15-00681-t001:** Characteristic features of HLVd.

Features	Characteristics
Structure of mature viroid RNA	Rod-like
Central conserved region (CCR)	Present
Terminal conserved hairpin (TCH)	Present
Self-cleaving activity (i.e., ribozyme)	absent
Replication site in the host	Nucleolus
Mode of Replication	Asymmetric rolling circle
Enzymes involved in replication	DNA-dependent RNA polymerase II

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
