# Peer review of "Hop Latent Viroid: A Hidden Threat to the Cannabis Industry"

_viruses, 2023, doi:10.3390/v15030681_

Round 1
Reviewer 1 Report
This review paper summarizes the work done on hop latent viroid in cannabis. It is organized in chapters of general nature introducing first the concept of viroids and specific viroids infecting cannabis and hop as cultivated plant similar to cannabis. More specific chapters follow describing relevant topics from the first reports of HLVd in cannabis to the sanitation measures and suggestions on the control strategies. The concept is ok and the review is timely considering the raising importance of cannabis as a high cash crop in the North America and Europe. Perhaps, some chapters could be shortened (some sentences on the effect of point mutations, part on the hop viroids and their effects) because it takes away the focus from the viroids in cannabis, especially if the target audience is in the cannabis industry. There is a number of technical errors (language, imprecisions) to be solved as marked in the uploaded edited pdf.
Nonetheless, these are not major issues and the manuscript can be prepared for publication relatively fast.

Author Response
Reviewer 1:
This review paper summarizes the work done on hop latent viroid in cannabis. It is organized in chapters of general nature introducing first the concept of viroids and specific viroids infecting cannabis and hop as cultivated plant similar to cannabis. More specific chapters follow describing relevant topics from the first reports of HLVd in cannabis to the sanitation measures and suggestions on the control strategies. The concept is ok and the review is timely considering the raising importance of cannabis as a high cash crop in the North America and Europe. Perhaps, some chapters could be shortened (some sentences on the effect of point mutations, part on the hop viroids and their effects) because it takes away the focus from the viroids in cannabis, especially if the target audience is in the cannabis industry. There is a number of technical errors (language, imprecisions) to be solved as marked in the uploaded edited pdf.
Nonetheless, these are not major issues and the manuscript can be prepared for publication relatively fast.
Response: Thank you for reviewing our manuscript and encouraging words. We made all the changes as suggested in .pdf. All these changes are in red color font. However, we did not shorten certain chapters because we believe it will help academicians and industries who want to know more about HLVd adaptation and hop viroids that have the potential to infect cannabis.
Reviewer 2 Report
The manuscript by Adkar-Purushothama et al. is a well-documented review article on Hop latent viroid (HLVd) and contains a lot of valuable information to both scientific community and farmers.
The manuscript could be improved by paying attention to the following minor issues or errors:
The readers would be interested in the following questions: Are there any cultivars resistant to HLVd? Do susceptible cultivars outnumber resistant ones?
What are we supposed to be seeing here in parentheses (line #174 and #179).
Some typos or simple errors were found in line #127, #132, #147, #315, #333, #359, #369, and #375. Maybe, there are more. So, it is recommended that the manuscript be carefully re-examined.
It seems that something is missing in the sentence the lines #383 and #384 (‘If so, what are …. on the cannabinoids.’).
There is an error or inconsistency in the description of references, for example ref #1 (line #405) and ref #6 (line #413).
Author Response
Q1: The readers would be interested in the following questions: Are there any cultivars resistant to HLVd? Do susceptible cultivars outnumber resistant ones?
Response: This is a precious suggestion. To the best of our knowledge, there are no HLVd resistance cultivars of cannabis. We included this statement in L393.
Q2: What are we supposed to be seeing here in parentheses (line #174 and #179).
Response: These are the references for the website-based statements. Since there are no guidelines to cite website information, we gave the link next to the statement in the main text.
Q3: Some typos or simple errors were found in line #127, #132, #147, #315, #333, #359, #369, and #375. Maybe, there are more. So, it is recommended that the manuscript be carefully re-examined.
Response: All errors are corrected.
Q4: It seems that something is missing in the sentence the lines #383 and #384 ('If so, what are …. on the cannabinoids.').
Response: We rewrote the sentence: "If so, what are the HLVd sequence variant-specific disease symptoms on the susceptible cannabis plant and HLVd variant specific effect on the cannabinoids." (L383-385).
Q5: There is an error or inconsistency in the description of references, for example ref #1 (line #405) and ref #6 (line #413).
Response: All the references are corrected in the revised manuscript.